# Altered Gut Structure and Anti-Bacterial Defense in Adult Mice Treated with Antibiotics during Early Life

**DOI:** 10.3390/antibiotics11020267

**Published:** 2022-02-18

**Authors:** Tânia Martins Garcia, Manon van Roest, Jacqueline L. M. Vermeulen, Sander Meisner, Jan Koster, Manon E. Wildenberg, Ruurd M. van Elburg, Vanesa Muncan, Ingrid B. Renes

**Affiliations:** 1Department of Gastroenterology and Hepatology, Tytgat Institute for Intestinal and Liver Research, Amsterdam UMC, AGEM, University of Amsterdam, 1105 BK Amsterdam, The Netherlands; t.martinsgarcia@amsterdamumc.nl (T.M.G.); m.vanroest@amsterdamumc.nl (M.v.R.); j.l.vermeulen@amsterdamumc.nl (J.L.M.V.); s.meisner@amsterdamumc.nl (S.M.); m.e.wildenberg@amsterdamumc.nl (M.E.W.); 2Laboratory for Experimental Oncology and Radiobiology, Center for Experimental and Molecular Medicine, Amsterdam University Medical Centers, University of Amsterdam, Cancer Center Amsterdam, 1105 AZ Amsterdam, The Netherlands; jankoster@amsterdamumc.nl; 3Department of Pediatrics, Amsterdam UMC, University of Amsterdam, 1105 AZ Amsterdam, The Netherlands; rm.vanelburg@amsterdamumc.nl (R.M.v.E.); ingrid.renes@danone.com (I.B.R.); 4Danone Nutricia Research, 3584 CT Utrecht, The Netherlands

**Keywords:** antibiotics, early life, long-term, small intestine, bacterial defense

## Abstract

The association between prolonged antibiotic (AB) use in neonates and increased incidence of later life diseases is not yet fully understood. AB treatment in early life alters intestinal epithelial cell composition, functioning, and maturation, which could be the basis for later life health effects. Here, we investigated whether AB-induced changes in the neonatal gut persisted up to adulthood and whether early life AB had additional long-term consequences for gut functioning. Mice received AB orally from postnatal day 10 to 20. Intestinal morphology, permeability, and gene and protein expression at 8 weeks were analyzed. Our data showed that the majority of the early life AB-induced gut effects did not persist into adulthood, yet early life AB did impact later life gut functioning. Specifically, the proximal small intestine (SI) of adult mice treated with AB in early life was characterized by hyperproliferative crypts, increased number of Paneth cells, and alterations in enteroendocrine cell-specific gene expression profiles. The distal SI of adult mice displayed a reduced expression of antibacterial defense markers. Together, our results suggest that early life AB leads to structural and physiological changes in the adult gut, which may contribute to disease development when homeostatic conditions are under challenge.

## 1. Introduction

Early life is a crucial moment in the development of mammals. Events that occur during the first weeks or months after birth can imprint long-lasting effects [1,2,3,4]. Among the external factors that can affect long-term health are early life antibiotics (AB). AB are commonly prescribed to neonates and children [5,6,7] due to the high incidence of respiratory infections in pediatric patients [8]. Furthermore, early life AB are used to treat gastrointestinal, skin, and ear infections. Many neonates also receive AB because of suspected early onset sepsis, especially after preterm birth [9]. However, in most cases, bacterial infection is not culture proven in uninfected infants [10]. Especially in the case of premature neonates, AB are used to treat sepsis or to prevent infection when newborn or maternal risk factors are present. Among the different classes of AB, the most used during early life are β-lactams (amoxicillin, ampicillin), aminoglycosides (gentamycin, amikacin), glycopeptides (vancomycin, teicoplanin), and nitroimidazoles (metronidazole).

In recent years, prolonged treatment with early life AB in humans has been linked with several pathologies later in life, such as obesity [11,12,13,14], diabetes [15,16], and inflammatory bowel diseases (IBD) [17,18,19]. As the gut is home to trillions of commensal bacteria, essential to the balanced functioning of an organism, it is not surprising that the disruption of the microbiome by AB can lead to gut-related pathologies. In preclinical mouse models, it has been shown that AB treatment in early life leads to changes in both gut microbiome and intestinal immunity that persist into adulthood, long after the treatment has stopped [20,21]. Moreover, the maturation of the intestinal epithelium is only completed after birth and alterations during this vulnerable period can disturb the normal maturation process and might lead to long-term consequences. We have recently demonstrated that antibiotic treatment during early life can directly affect the intestinal epithelial cell (IEC) composition and functioning, independent of the effects on the microbiome [22]. Pups treated with AB between P10 and P20 showed accelerated intestinal maturation, as demonstrated by the decrease in intestinal permeability and the disappearance of specialized vacuolated cells, characteristic of the neonatal period. Furthermore, in vivo AB treatment of neonatal pups and in vitro AB treatment of fetal intestinal organoids induced the expression of adult specific brush-border enzymes, in both proximal and distal SI, and enteroendocrine cell (EEC) markers, particularly in the proximal SI [22]. Finally, AB-treated pups displayed higher expression of innate defense markers, especially in the distal SI [22]. Previous studies have also shown that adult mice that received AB during weaning are more susceptible to intestinal colitis and colon cancer when challenged with DSS and/or azoxymethane [23,24,25,26]. Altogether, these data suggest that the direct and indirect effects of AB on the immature neonatal gut can have long-lasting consequences and predispose to gut related diseases. Yet, the long-term effects of early life AB observed in the homeostatic gut, i.e., without any challenge, have never been described. Therefore, it is essential to identify the direct impact of early life AB on gut development and later life gut functioning in order to better target gut health and prevent diseases.

In the current study, we investigated whether the epithelial changes resulting from the direct effect of AB early in life, i.e., before and during weaning, persist in the adult intestine. Mice were treated between postnatal days 10 and 20 with amoxicillin, vancomycin, and metronidazole. At 8 weeks of age small intestinal morphology, permeability, and proliferation were analyzed. The expression of small intestinal epithelial genes and proteins previously shown to be directly affected by AB in early life was investigated. Furthermore, by analyzing the global expression of adult IECs we identified additional long-term consequences of AB in early life.

## 2. Results

### 2.1. Adult Mice Treated with Early Life Antibiotics Show Altered Small Intestine Morphology

To investigate the long-term effects of early life AB, we treated mice daily by oral gavage, from postnatal day (P)10 to P20, with a mix of AB comprising the most commonly used classes in neonates: amoxicillin, vancomycin, and metronidazole (Figure 1A). No aminoglycoside AB was used as this class is not absorbed by the gastrointestinal epithelial cells. At P20, AB treatment ended and mice were weaned (Figure 1A). As of week 3, bodyweight was similar between AB-treated mice and control mice, which received PBS (Figure 1B). We previously showed that the intestinal permeability of P20 pups treated with the same early life AB mix was reduced compared to control pups [22]. To evaluate whether this difference in intestinal permeability persists into adulthood, we orally administered FITC-dextran to 8-week-old mice, that had been treated with AB in early life, and quantified its concentration in serum four hours later (Figure 1C). The FITC-dextran serum concentration was identical in AB-treated and control mice indicating that intestinal permeability normalized over time after AB treatment (Figure 1D). Still, analysis of the small intestine (SI) showed that both its weight and length were significantly higher in mice that received early life AB compared to control mice (Figure 1E,F). Finally, the liver weight of AB-treated mice was significantly lower than control mice, while spleen weight was not different between both groups (Figure 1G,H). Although the differences in intestinal permeability immediately after AB treatment are no longer present in adulthood, the increased weight and length of the SI suggest that early life AB have long-lasting effects on the intestine.

### 2.2. Early Life Antibiotics Induce Hyperproliferative Crypts in Adult Proximal Small Intestine

The epithelial cells of the proximal and distal SI have different genetic profiles that support specific regional nutrient absorption and digestion functions, as well as different immune defense requirements [27,28,29,30]. We previously showed that these two SI regions are differently affected by early life AB treatment at P20 [22]; thus in the present study, we analyzed adult proximal and distal SI separately. Assessment of SI histology revealed no major morphological differences (Figure 2A). A trend towards shorter villi and significantly longer crypts was observed in the proximal SI of AB-treated mice compared to control mice, while distal SI showed similar villus lengths and crypt depths in the two groups (Figure 2B,C). As longer crypts suggest increased proliferation, we performed staining for the mitotic marker phosphorylated histone H3 (PHH3) (Figure 2D). Proximal SI of AB-treated mice displayed a higher number of PHHH3-stained cells compared to control mice and in the distal SI of these mice only, a trend toward increased PHHH3-stained cells was noted (Figure 2E). These results indicate that early life AB induce hyperproliferative crypts in the proximal SI.

### 2.3. The Majority of the Direct Effects Induced by Antibiotics on the Neonatal Small Intestine Do Not Persist into Adulthood

The postnatal maturation of the mouse intestinal epithelium is characterized by several changes in its morphology and function. For example, from P14 to P28, the SI brush-border starts to express adult-specific enzymes, while the number of Paneth cells expands in the newly developed crypts [31]. We have shown that AB treatment of mice between P10 and P20 led to upregulation of the adult brush-border enzymes and Paneth cells at P20 [22]. We also demonstrated that these changes resulted from the direct effect of AB on IECs [22]. Thus, here we set out to determine whether these effects were still present in adult mice that received the same AB treatment in early life. In contrast with the observations at P20, whole tissue qRT-PCR analysis on the adult proximal SI showed similar relative expression of the brush-border enzymes Sis and Arg2 between AB-treated mice and control adult mice (Figure 3A). However, the relative expression of the Paneth cell marker lysozyme-1 (Lyz1) was higher in AB-treated adult mice compared to control mice (Figure 3A). Immunohistochemistry of Lyz1 also revealed a significantly higher number of Paneth cells in the adult proximal SI after early life AB treatment (Figure 3B). In the distal SI, Arg2 showed a trend towards an increased relative expression after AB treatment, but Sis and Lyz1 were similarly expressed between the two groups (Figure 3C). In accordance, immunohistochemistry of Lyz1 showed a similar amount of Paneth cells in the distal SI in both groups (Figure 3D).

The upregulation of EEC markers observed after AB treatment in proximal SI at P20 did not persist into adulthood, as measured by qRT-PCR of 8-week-old SI whole tissue (Figure 3E). Only the relative expression of secretin (Sct) remained significantly higher, while cholecystokinin (Cck), which relative expression was not changed by AB at P20, was significantly lower in AB-treated adult mice compared to control mice (Figure 3E).

Overall, the adult SI recovered from the direct effects of early life AB treatment, with the exception of the greater number of Paneth cells, the higher relative expression of Sct, and the lower relative expression of Cck in the proximal SI.

### 2.4. Genome-Wide Gene Expression Analysis Reveals Modest Differences in Epithelial Cells of Adult Mice Treated with Antibiotics in Early Life

Early life AB treatment broadly depletes the commensal microbiota of the developing gut, which affects the IECs gene expression profile in early life [13,20,23,24,32]. The unbalanced microbiota expansion that follows after the AB treatment at P20 can also affect IEC-specific gene expression. To examine whether the AB-induced changes in IEC gene expression persist into adulthood, we next performed genome-wide gene expression analysis on mRNA from proximal and distal SI epithelial cells of 8-week-old mice that had received AB or PBS in early life (Figure 4A). Principal component analysis (PCA) showed a modest separation between AB-treated and control proximal SI epithelial cells along the second component (PC2 23.1%) (Figure 4B). This separation was not found in the analysis of distal SI epithelial cells (Figure 4C). Nevertheless, differential gene expression analysis showed a similar number of at least 2-fold upregulated or downregulated genes in AB-treated epithelial cells from both regions (Figure 4D,E). Specifically, 92 genes in proximal SI and 137 genes in distal SI epithelial cells were upregulated. Next, we grouped these differently expressed genes into specific functions. This resulted in a single function being clearly affected specifically in the distal SI: “antibacterial defense”. These data showed that, of the few differences in adult IECs gene expression caused by AB treatment in early life, “antibacterial defense” seems to be the only function affected especially in the distal SI.

### 2.5. Intestinal Antibacterial Defense Is Reduced in Distal Small Intestine of Adult Mice That Received Antibiotics in Early Life

The identified genes involved in antibacterial defense were downregulated in the distal SI epithelial cells of AB-treated mice compared to control mice at 8 weeks of age (Figure 5A). This is in contrast with our findings at P20 which showed upregulated innate defense markers, including Reg3 lectins, in AB-treated pups [22]. Thus, we then compared the gene expression of these antibacterial defense genes between the distal SI of P20 pups and adult mice. We found that while control adult mice showed a sharp increase in the expression of antibacterial defense genes compared to control neonatal pups (Figure 5B), AB-treated adult mice showed similar expression levels of these genes compared to AB-treated neonatal pups (Figure 5C). This indicates that the developmental upregulation in gene expression of antibacterial defense markers taking place in the distal SI of control mice does not occur in AB-treated mice.

Colitis is one of the pathologies that has been associated with the use of antibiotics during early life [17,18,19]. We examined in more detail the expression of regenerating islet-derived protein 3 lectins (Reg3) and serum amyloid A proteins (Saa), as these have been described to be inversely correlated with the development of colitis [33,34]. An independent experiment was performed with exactly the same conditions and whole distal SI was analyzed by qRT-PCR, confirming these results, except for Saa3 (Figure 5D). Finally, visualization of goblet cells, which are key players in gut barrier defense, by staining of mucins, showed a similar number of this secretory cell type between antibiotic-treated and control mice (Figure 5E). Overall, these results showed that early life AB treatment limits intestinal antibacterial defense in the distal small intestine by preventing the normal developmental pattern of these defense genes from weaning into adulthood.

## 3. Discussions

In this study, we found that pre-weaning treatment of young mice with AB results in permanent higher SI weight and increased SI length later in life (Figure 1). This might be partly explained by the crypt hyperproliferation observed in the proximal SI of AB-treated mice, supported by the higher amount of Paneth cells, and the consequent increase in crypt depth (Figure 2 and Figure 3). Certain bacterial strains can influence crypt proliferation. Specifically, Gram-positive bacteria and associated short-chain fatty acids (SCFAs) production are known to increase IECs turnover [35]. Lactic acid-producing bacteria have also been shown to stimulate intestinal stem cell proliferation [36]. Examination of the intestinal bacterial composition before and after AB treatment, as well as later in life, could provide additional insights into whether the observed increase in SI weight, length, and proliferation is due to microbial signals. Nevertheless, the fact that morphological differences are only evident in the poorly microbial-populated proximal SI compared to the distal SI indicates that these differences might be independent of microbiota.

The strongly reduced intestinal permeability measured at P20 [22] was not observed in adulthood (Figure 1). Decreased permeability, and thus increased barrier function, is one of the hallmarks of intestinal epithelial maturation that in mice occurs between the second and fourth week after birth [37]. The similarity in intestinal permeability observed in adulthood between AB-treated and control mice further demonstrates that its reduction at P20 was a result of the accelerated maturation caused by AB treatment and, thus, an early life effect.

Although the increase in expression of adult brush-border genes was a short-term effect of early life AB in both SI regions [22], at 8 weeks only Arg2 remained increased in the distal SI of AB-treated mice (Figure 3). Bacteria have been linked to Arg2 expression [38,39], which could explain its specific increase in distal SI at 8 weeks of age. Previous studies have demonstrated that altered microbiota composition induced by AB during early life can persist into adulthood [12,20,24,40]. Therefore, it may well be that the microbiota of 8-week-old AB-treated mice is associated with changes in epithelial expression, as we observed for Arg2.

At P20, early life AB treatment induced expression of EEC markers in the proximal SI of both neonatal pups and fetal intestinal organoids, demonstrating that the increase in EEC gene expression and number of EEC is a direct effect of AB on IECs [22]. In AB-treated adult mice, only the expression of Sct remained upregulated, while Cck gene expression, which was not changed at P20, was downregulated (Figure 3). EEC secrete hormones that regulate digestion, epithelial barrier integrity, and mucosal immunity [41]. Inappropriate release of EEC hormones can be involved in the development of obesity and diabetes [42,43,44]. For example, the marker Sct encodes the hormone secretin which induces insulin release. When insulin levels are persistently high, cells develop insulin resistance, an early sign of type 2 diabetes. In our study, we observed a higher relative gene expression of Sct in AB-treated mice, which could contribute to the AB-linked development of type 2 diabetes [45,46]. However, it is not known whether the increase in Sct relative expression causes continuous increased insulin release and consequently type 2 diabetes development. Moreover, the downregulated Cck expression in AB-treated adult mice might have detrimental effects on bacterial growth and intestinal inflammatory conditions. Specifically, decreased Cck might limit the capacity of the gut to prevent bacterial overgrowth and lead to increased translocation of bacteria into the intestinal mucosa [47], which can stimulate pro-inflammatory conditions and exacerbate intestinal inflammation.

Antibacterial defense in the distal SI was the only function still profoundly affected at 8 weeks of age (Figure 5). The expression of antibacterial genes increases from the neonatal stage to adulthood, but AB blocked this increase. The distal SI epithelium is not capable of recovering from the AB treatment and the antibacterial defense gene expression levels in control adult mice are not reached in AB adult mice. Thus, not only does the negative impact on the gut microbiome and intestinal immunity caused by early life AB persist in adulthood as described previously [20,21] but also epithelial defense response is affected into adulthood. Consequently, the distal SI of adult mice that received early life AB is less protected at homeostatic conditions, which will leave it more vulnerable to bacterial infections and dysbiosis, thereby increasing the susceptibility to disease. Furthermore, Reg3 lectins, which were among the downregulated antibacterial defense markers, have bactericidal activity against Gram-positive bacteria, controlling their presence in the gut and contributing to the microbiome composition [48,49,50,51], and lower Reg3 expression is also associated with a higher risk of colitis in mice [34]. Intestinal chronic inflammation pathologies, such as ulcerative colitis and Crohn’s disease, result from inadequate response to intestinal microbiota, including commensal bacteria. Decreased intestinal antibacterial defense due to early life AB treatment can lead to dysbiosis, promoting the onset of inflammatory conditions. Ultimately, bacteria may more easily translocate from the lumen to extraintestinal sites and contribute to the exacerbation of inflammation symptoms. Indeed, adult mice that received AB in early life showed increased susceptibility to DSS-induced colitis, colon cancer, obesity, and diabetes [11,16,23,40]. In addition, a recent systematic review of clinical studies in children showed strong evidence for an association between early life AB and both IBD and celiac disease in later childhood [52]. IBD is a multifactorial disease affecting the gut immune system, epithelium, and microbiome. Previously, it was demonstrated that early life AB causes changes in the microbiome that persist into adulthood, accompanied by a negative impact on the immune system [20,21]. Our present study shows that the epithelial defense response is yet another aspect of the long-lasting effects of AB in early life. Further studies are required to demonstrate the association between reduced epithelial antibacterial defense and increased predisposition to IBD, and to understand the underlying mechanisms.

In conclusion, early life AB life leads to intestinal structural differences in adulthood, hyperproliferation of the proximal SI, increased Sct and decreased Cck expression in the proximal SI, and impaired antibacterial defense in the distal SI, which may contribute to the increased incidence of gut-related diseases when homeostatic conditions are challenged. Validation of these findings in the human small intestine will help to develop targeted and personalized strategies, for neonates and children that receive prolonged antibiotic treatment, for example using prebiotics, probiotics, or synbiotics.

## 4. Materials and Methods

### 4.1. In Vivo Studies

This study was conducted in accordance with institutional guidelines for the care and use of laboratory animals established by the Animal Ethics Committee of the University of Amsterdam, and all animal procedures related to the purpose of the research were approved under the Ethical license of the national competent authority, securing full compliance the European Directive 2010/63/EU for the use of animals for scientific purposes.

Six pregnant 8 weeks old C57Bl/6J females were obtained from Charles River and were allowed to adapt to the new environment for 1 week. Pregnant females were individually housed and received an AIN-93G diet (Triple A Trading/Altromin, Tiel, Netherlands). Mice were kept in innovive Universal Euro II Type Long disposable mice cages (522.6 cm^2^ floor space, 5653.5 cm^3^ living space, 12.7 cm height), with corncob bedding (type 1/8 corncob, Innovive, San Diego, CA, USA) and carton house (15 cm × 7 cm × 7 cm, 25 Gky irradiated, Tecnilab Des Res, Someren, the Netherlands), with tissue as nesting material (Facial Tissue Extra Soft, PK100ST, King Nederland Tork, Tiel, Netherlands). Cages were cleaned every two weeks. Lights were on at 7am and off at 7pm and all procedures occurred during the light phase. The temperature was kept between 20 °C and 24 °C and humidity between 45% and 65%. Food and water (acidified to 2.5–3.0 pH, Aquavive^®^, M-WB-300A) were given ad libitum. Pups were monitored daily from outside the cage for deviations in behavior and physical health, weighted every other day from P10 onwards, and kept with the mothers throughout the experiment. At P10, two experimental groups were randomly defined: treatment group (3 litters, 4 pups per litter, 6 females, and 6 males) received daily oral gavage, using a plastic feeding tube (FTP2225, 22 G × 25 mm, Instech Solomon, Plymouth Meeting, PA, USA) of 30 µL of AB (25 mg/kg/day amoxicillin (Amsterdam UMC, AMC pharmacy, Amsterdam, The Netherlands), 50 mg/kg/day metronidazole (Amsterdam UMC, AMC pharmacy, Amsterdam, The Netherlands), and 50 mg/kg/day vancomycin (Sigma-Aldrich, Amsterdam, Netherlands); control group (3 litters, 4 pups per litter (except 1 litter which had 3 pups), 5 females and 6 males) received daily oral gavage of 30 µL of PBS. Mice were physically restrained by scruffing (single hand method) during oral gavage. AB or PBS were consistently given during the light period, always at the same time period of the day, nonblinded. Measurements were performed by a different person in a blinded fashion. For oral gavage, pups were separated from the mother all at once and placed back all at once as well, to correct for differences in maternal care. On P21, pups were weaned and the 6 litters were distributed into 4 different cages: control females (*n* = 5), control males (*n* = 6), antibiotic-treated females (*n* = 6), and antibiotic-treated males (*n* = 6). Mice were monitored daily from outside the cage for deviations in behavior and physical health and weighted every other day. At 8 weeks, mice were fasted and after 3 h, 250 µL of 60 mg/100 gr weight FITC-dextran 4 kDa (Sigma-Aldrich) diluted in PBS were given via oral gavage (plastic feeding tube, Instech Solomon, FTP2225, 22 G × 25 mm) to all the mice, to assess intestinal permeability. Mice were physically restrained by scruffing (single hand method) during oral gavage. After 4 h, mice were euthanized (one individual at a time) by 100% CO_2_ and 4% isoflurane exposure (20% volume/min) inside a small container. Immediately after, blood was collected by heart puncture in MiniCollect^®^ Z Serum Sep Clot tubes (Greiner, Alphen aan den Rijn, Netherlands). After 30 min incubation on ice, in the dark, blood was centrifuged and serum was collected and kept at −80 °C.

### 4.2. FITC-Dextran In Vivo Permeability Assay

Standard samples were obtained by 2-fold serial dilution of 1 mg/mL FITC-dextran in blood serum. The fluorescence signals of the serum samples were recorded with an excitation wavelength of 485 nm and emission wavelength of 520 nm and compared with the standard curve values. The amount of FITC-dextran in serum samples was calculated in ng/mL.

### 4.3. Immunostaining

Tissue was flushed with PBS, fixed overnight in 4% formaldehyde, embedded in paraffin, and sectioned. Sections were deparaffinized with xylene and gradually rehydrated in ethanol. After blocking the endogenous peroxidase (0.01% H_2_O_2_ in methanol), slides were boiled in 0.01 M sodium citrate buffer (pH 6) for 10 min at 120 °C in an autoclave for antigen retrieval. Slides were blocked for 30 min at room temperature in PBS with 1% bovine serum albumin and 0.1% Triton-X-100. Then, slides were incubated overnight with primary antibody diluted in the blocking buffer. Slides were washed with PBS and secondary antibody diluted in blocking buffer was added for 30 min at room temperature. Antibody binding was visualized by adding chromagene substrate diaminobenzedine (Sigma-Aldrich, Amsterdam, The Netherlands), sections were counterstained using haematoxillin (Sigma-Aldrich, Amsterdam, The Netherlands) and slides were dehydrated and mounted with entellan.

For Alcian blue and periodic acid (AB-PAS) staining, deparaffinized sections were stained with Alcian blue (Sigma, A3157) for 20 min at room temperature, washed under running tap water for 5 min and then in bidistilled water before incubating with freshly prepared 0.1% periodic acid (Sigma, P0430). Slides were then washed once more under running tap water for 10 min and stained with Schiff’s reagent (J62171.AP, VwR International, Amsterdam, The Netherlands) also for 10 min. Slides were washed under running tap water for 5 min and in bidistilled water for another 5 min. Finally, sections were counterstained with hematoxylin (Sigma) and slides were dehydrated and mounted with entellan. Sections were examined using brightfield microscope Olympus BX51 and analysis was performed with blinded slides. Per mouse, the length of at least 10 villi and the depth of at least 10 crypts were measured. For AB-PAS analysis, at least 20 villi and 40 crypts were quantified. To quantify the number of phosphor-histone H3 and LYZ1 stained cells per crypt, at least 35 crypts in the proximal SI and at least 40 crypts in the distal SI were quantified, per mouse.

Phospho-Histone H3-rabbit polyclonal anti-phospho-histone H3 (1:200, ThermoFischer, PA5-17869).

LYZ1-rabbit polyclonal anti-lysozyme (1:2000, DAKO, A0099).

### 4.4. Epithelial Cells FACS-Sorting

The small intestine of 8-week-old mice was cut open and proximal and distal parts were separated, cut into pieces, and washed with ice-cold PBS. Crypts were dissociated after incubation with 2 mM EDTA (Merck/VWR) for 30 min at 4 °C and filtered through a 70 µm cell strainer (BD/VWR). Single cells were obtained by incubating crypts with TrypLE Express (Invitrogen). Cells were kept in PBS 2% FCS Rho-kinase inhibitor and RNase inhibitor (Fermentas/Thermo Fisher Scientific) solution and stained with EpCAM-FITC antibody (1:50, 324204, BioLegend, San Diego, CA, USA) for 30 min on ice.

### 4.5. RNA Isolation and qRT-PCR

For transcriptome profiling, RNA was extracted from EpCAM-positive cells using the phenol–chloroform method. RNA quality was measured on an Agilent 2100 Bioanalyzer.

For qRT-PCR, RNA from whole-tissue tissue was isolated using the Bioline ISOLATE II RNA Mini kit (BIO-52073, Bioline) according to manufacturers’ instructions. Then, 1 µg of RNA was transcribed using Revertaid reverse transcriptase according to the manufacturer’s protocol (Fermentas, Vilnius, Lithuania). Quantitative RT-PCR was performed on a BioRad iCycler using sensifast SYBR No-ROX Kit (GC-biotech Bio-98020) according to the manufacturer’s protocol. The two most stable reference genes were determined using GeNorm and their geometric mean was used to calculate the relative expression of genes of interest: for whole-tissue qRT-PCR, ribosomal protein L4 (Rpl4) and peptidylpropyl isomerase B (Ppib). Relative gene expression was calculated using N0 values obtained by LinRegPCR analysis. Primers were previously validated using melting curve analyses and gel electrophoresis of PCR products.
Rpl4: FW-CCTTCTCCTCTCCCCGTCA ; RV-GCATAGGGCTGTCTGTTGTTTPpib: FW-GCCAACGATAAGAAGAAGGGA; RV-TCCAAAGAGTCCAAAGACGACSis: FW-TGCCTGCTGTGGAAGAAGTAA; RV-CAGCCACGCTCTTCACATTTArg2: FW-TAGGGTAATCCCCTCCCTGC; RV-AGCAAGCCAGCTTCTCGAATLyz: FW-GGATGGCTACCGTGGTGTCAAGC; RV-TCCCATAGTCGGTGCTTCGGTCReg3β: FW-TGGGAATGGAGTAACAAT; RV-GGCAACTTCACCTCACATReg3γ: FW-CCATCTTCACGTAGCAGC; RV-CAAGATGTCCTGAGGGCGip: FW-AACTGTTGGCTAGGGGACAC; RV-TGATGAAAGTCCCCTCTGCGGcg: FW-CTTCCCAGAAGAAGTCGCCA; RV-GTGACTGGCACGAGATGTTGPyy: FW-ACGGTCGCAATGCTGCTAAT; RV-GCTGCGGGGACATCTCTTTTTSst: FW-GACCTGCGACTAGACTGACC; RV-CCAGTTCCTGTTTCCCGGTGSct: FW-GACCCCAAGACACTCAGACG; RV-TTTTCTGTGTCCTGCTCGCTCck: FW-GAAGAGCGGCGTATGTCTGT; RV-CCAGAAGGAGCTTTGCGGAChgA: FW-GTCTCCAGACACTCAGGGCT; RV-ATGACAAAAGGGGACACCAASaa1: FW-GGTCTTCTGCTCCCTGCTC; RV-AGCAGCATCATAGTTCCCCCSaa2: FW-CAGCCTGGTCTTCTGCTCC; RV-CACATGTCTCCAGCCCCTTGSaa3: FW-AGTAGGCTCGCCACATGTCT; RV-TCCATTGCCATCATTCTTTG

### 4.6. Transcriptome Profiling

For transcriptome profiling, 400 ng RNA was amplified and labeled using the 3′ IVT Pico Kit Affymetrix RNA Amplification Kit (Nugene) according to the manufacturer’s protocol. Microarray analysis of mouse EpCAM-positive cells was performed using the Affymetrix Clariom^®^ S 8-Array HT Plate according to the standard protocols of the Dutch Genomics Service and Support Provider (MAD, Science Park, University of Amsterdam, Netherlands). The data were normalized using Expression Console 1.4.1.46 and uploaded to R2: Genomics Analysis and Visualization Platform (http://hgserver1.amc.nl/, accessed on 28 August 2019). Microarray results were analyzed using R2 software version 3.4.3. Differentially expressed genes were selected based on fold change (≥2) in comparison to the control group.

### 4.7. Software

nQuery 7.0 was used for sample-size calculations and ImageJ for villi length and crypt depth measurement. Microarray data was analyzed by Transcriptome Analyses Controle (TAC) and R2. GeNorm was used for identification of most stable reference genes for quantitative Real-time PCR analyses performed by LinRegPCR. GraphPad Prism 8 was used for statistical analyses and graph creation.

### 4.8. Statistical Analysis

Sample size was calculated using nQuery and based on the effect size of maturation studies, using a two-group *t*-test of equal n’s, with a significance level (α) of 0.05 and power of 80%. There were no exclusions or drop-outs. Data were analyzed using GraphPad Prism 8 and presented as the mean ± standard deviation unless stated otherwise in the figure legends. Sample distribution was determined using D’Agostino and Pearson normality test. Sample numbers, experimental replicates, type of statistical analysis test, and *p* values are reported in the figure legends.

## Figures and Tables

**Figure 1 antibiotics-11-00267-f001:**
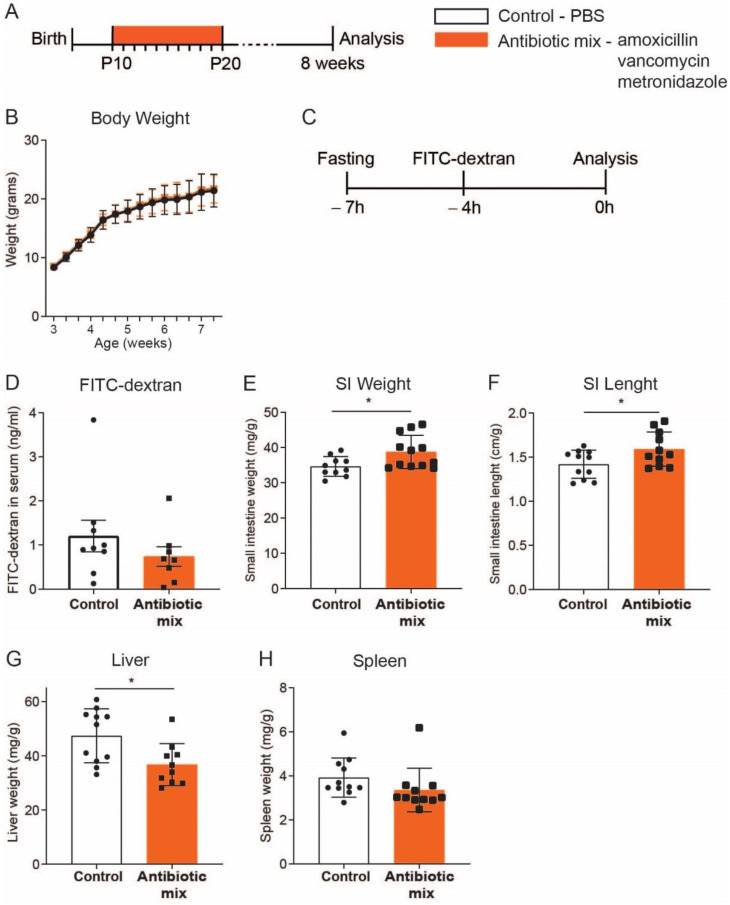
Growth, intestinal permeability, and macroscopic assessment of small intestine, liver, and spleen. (**A**) Experimental design of in vivo antibiotic treatment of pups between postnatal (P) day 10 and P20. Antibiotic mix: amoxicillin, metronidazole, and vancomycin. All analyses were carried out at 8 weeks of age. (**B**) Mice weight was measured three times per week after antibiotic mix treatment, *n* = 11 control mice, *n* = 12 AB-treated mice. (**C**,**D**) Permeability assay assessed by FITC-dextran concentration in serum 4 h after oral gavage, *n* = 9 control mice, *n* = 8 AB-treated mice. (**E**) Small intestine weight, relative to body weight, *n* = 10 control mice, *n* = 12 AB-treated mice. (**F**) Small intestine length, relative to body weight, *n* = 11 control mice, *n* = 11 AB-treated mice. (**G**) Liver weight, relative to body weight, *n* = 11 control mice, *n* = 10 AB-treated mice. (**H**) Spleen weight, relative to body weight, *n* = 11 control mice, *n* = 11 AB-treated mice. Statistical analysis was performed by two-way ANOVA test with Sidak’s multiple comparisons test (**B**) or two-tailed unpaired *t*-test (**D**–**H**). Error bars indicate mean ± SD. Levels of significance are indicated (* *p* < 0.05).

**Figure 2 antibiotics-11-00267-f002:**
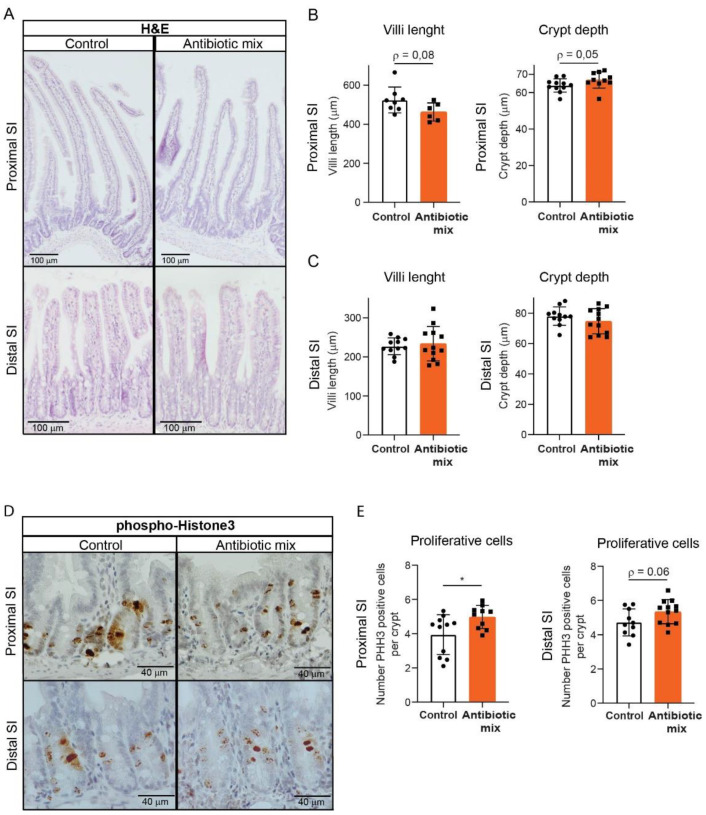
Hyperpoliferative crypts in adult mice treated with early life antibiotics. (**A**) H&E staining of proximal and distal small intestine. (Scale bars, 100 µm.) (**B**) Villus length and crypt depth in proximal small intestine. Villus: *n* = 8 control mice, *n* = 6 AB-treated mice; crypt: *n* = 11 control mice, *n* = 10 AB-treated mice. (**C**) Villus length and crypt depth in distal small intestine. Villus: *n* = 11 control mice, *n* = 12 AB-treated mice; crypt: *n* = 11 control mice, *n* = 12 AB-treated mice. (**D,E**) Immunohistochemistry of proliferation marker phosphorylated histone H3 (**D**) and quantifications of positive-stained cells (**E**) in proximal and distal small intestine. Proximal: *n* = 11 control mice, *n* = 10 AB-treated mice; distal: *n* = 10 control mice, *n* = 12 AB-treated mice. (Scale bars, 40 µm.) Statistical analysis was performed by Mann-Whitney test when data were not normally distributed as assessed by D’Agostino and Pearson normality test (**B**) or two-tailed unpaired *t*-test (**C**,**E**). Error bars indicate the mean ± SD. Levels of significance are indicated (* *p* < 0.05).

**Figure 3 antibiotics-11-00267-f003:**
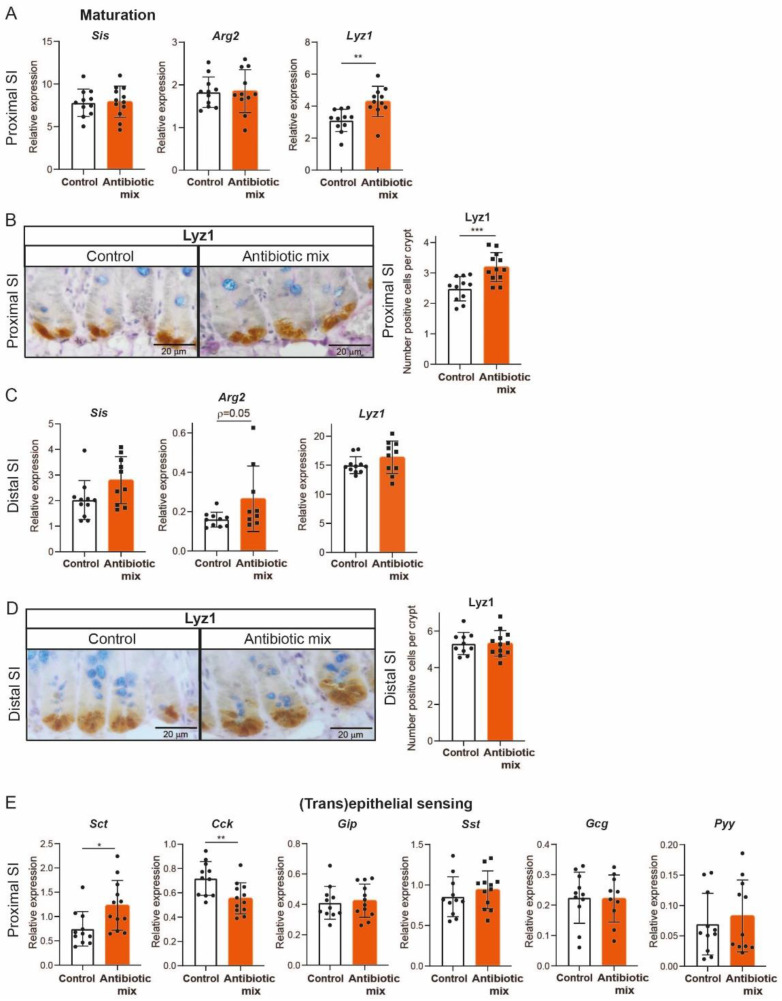
Antibiotics in early life lead to higher number of Paneth cells and few changes in enteroendocrine markers in the proximal small intestine. (**A**) Whole tissue real-time qPCR analysis of adult brush-border enzymes Sis and Arg2 and of Paneth cell marker Lyz1 in proximal small intestine. Relative expression to reference genes Rpl4 and Ppib. Sis: *n* = 11 control mice, *n* = 12 AB-treated mice; Arg2: *n* = 11 control mice, *n* = 11 AB-treated mice; Lyz1: *n* = 11 control mice, *n* = 10 AB-treated mice. (**B**) Immunohistochemistry of lysozyme-1 in proximal small intestine, *n* = 11 control mice, *n* = 12 AB-treated mice. (Scale bars, 20 µm.) (**C**) Whole tissue real-time qPCR analysis of adult brush-border enzymes Sis and Arg2 and of Paneth cell marker Lyz1 in distal small intestine. Relative expression to reference genes Rpl4 and Ppib. Sis: *n* = 11 control mice, *n* = 10 AB-treated mice; Arg2: *n* = 10 control mice, *n* = 9 AB-treated mice; Lyz1: *n* = 11 control mice, *n* = 10 AB-treated mice. (**D**) Immunohistochemistry of lysozyme-1 in distal small intestine, *n* = 10 control mice, *n* = 12 AB-treated mice (Scale bars, 20 µm.) (**E**) Whole tissue real-time qPCR analysis of enteroendocrine markers Sct, Cck, Gip, Sst, Gcg, and Pyy in proximal small intestine. Relative expression to reference genes Rpl4 and Ppib. Sct, Cck, Gip, Sst: *n* = 11 control mice, *n* = 12 AB-treated mice. Gcg: *n* = 11 control mice, *n* = 10 AB-treated mice. Pyy: *n* = 11 control mice, *n* = 11 AB-treated mice. Statistical analysis was performed by one-tailed unpaired *t*-test or by Mann–Whitney test when data were not normally distributed as assessed by D’Agostino and Pearson normality test (Sct). Error bars indicate the mean ± SD. Levels of significance are indicated (* *p* < 0.05, ** *p* < 0.01, *** *p* < 0.005).

**Figure 4 antibiotics-11-00267-f004:**
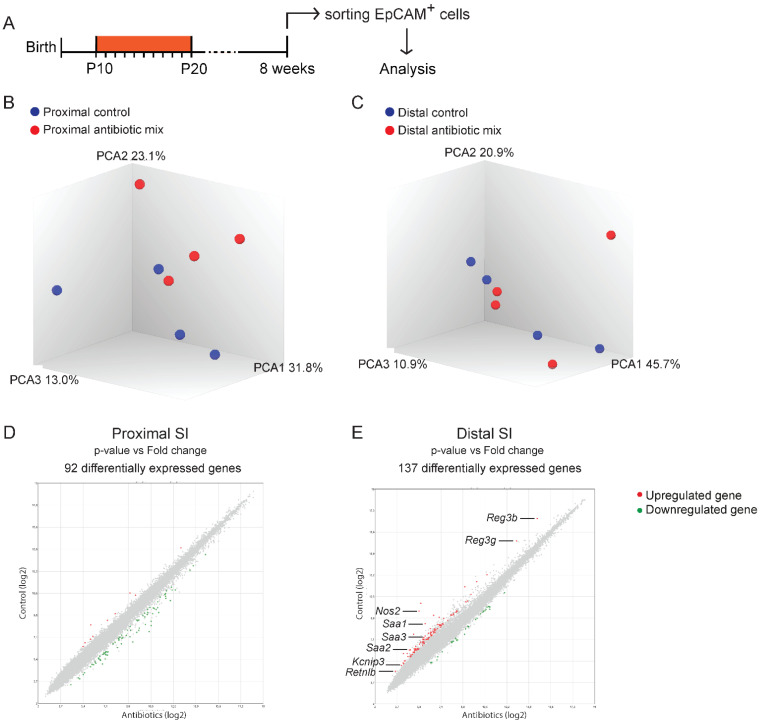
Genome-wide gene expression analysis of sorted intestinal epithelial cells. (**A**) Experimental design of genome-wide gene expression analysis of FACS-sorted intestinal epithelial cells of 8-week-old mice. (**B**,**C**) PCA analysis of sorted epithelial cells from 8-week-old proximal (**B**) and distal (**C**) small intestine treated with the antibiotic mix or PBS (control) in early life. (**D**,**E**) Volcano plots of microarray analysis showing genes differentially expressed between control and antibiotic-treated FACS-sorted intestinal epithelial cells of proximal (**D**) and distal (**E**) small intestine of 8-week-old mice. Green dots identify downregulated genes and red dots identify upregulated genes. Statistical analysis by ANOVA eBayes, *p* < 0.05 cut-off, *n* = 4 samples per group.

**Figure 5 antibiotics-11-00267-f005:**
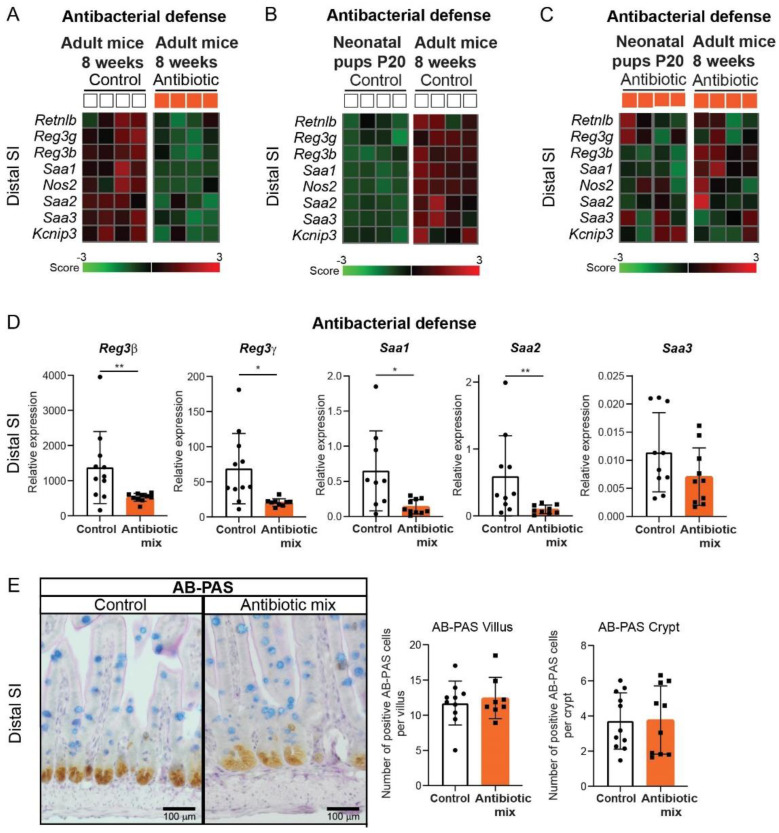
Reduced antibacterial defense in distal small intestine of adult mice treated with early life antibiotics. Differential gene expression analysis of sorted intestinal epithelial cells after antibiotic treatment. (**A**–**C**) Curated heatmaps of selected genes from top downregulated and top upregulated genes, based on biological interest and grouped according to function “antibacterial defense” in distal epithelial cells of control and AB-treated adult mice (**A**), control neonatal pups and adult mice (**B**), and AB-treated neonatal pups and adult mice (**C**). The colored bar represents the expression level from low (green) to high (red), *n* = 4 samples per group. (**D**) Whole tissue real-time qPCR analysis of antibacterial defense markers Reg3β, Reg3γ, Saa1, Saa2, and Saa3 in distal small intestine. Relative expression to reference genes Rpl4 and Ppib. Reg3β: *n* = 11 control mice, *n* = 10 AB-treated mice. Reg3γ: *n* = 11 control mice, *n* = 10 AB-treated mice. Saa1: *n* = 9 control mice, *n* = 10 AB-treated mice. Saa2: *n* = 10 control mice, *n* = 9 AB-treated mice. Saa3: *n* = 10 control mice, *n* = 10 AB-treated mice. (**E**) Staining and quantification of mucins in villus and crypts of distal small intestine. Villi: *n* = 10 control mice, *n* = 8 AB-treated mice; crypts: *n* = 11 control mice, *n* = 10 AB-treated mice. (Scale bars, 100 µm.) Statistical analysis by ANOVA eBayes, *p* < 0.05 cut-off (**A**–**C**) or by one-tailed unpaired *t*-test or Mann–Whitney test (Reg3β, Saa2), when data were not normally distributed as assessed by D’Agostino and Pearson normality test (Sct). Error bars indicate the mean ± SD. Levels of significance are indicated (* *p* < 0.05, ** *p* < 0.01).

## Data Availability

Microarray data are deposited in the Gene Expression Omnibus Database and available upon request.

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
