# Peer review of "Altered Gut Structure and Anti-Bacterial Defense in Adult Mice Treated with Antibiotics during Early Life"

_antibiotics, 2022, doi:10.3390/antibiotics11020267_

Round 1

Reviewer 1 Report

This work is a supplementary study to Garcia et al. 2021 (Cell Mol Gastroenterol Hepatol, PMID: 34102314) on the effects of early life antibiotic treatment on gut functions.  In this part of their study, they showed that the major of the effects induced by AB treatment from PND 10 to 20 on the neonatal SI did not persist in the adulthood. i.e.  at the age of 8 weeks, although this treatment was showed to induce hypeproliferative crypts in proximal SI and reduced expression of genes coding for antibacterial defences in the distal SI in adult mice. The study is interesting, and the study design is adequate. The paper is well-written.

Minor points.

  1.  Since the experiments were performed on both male and female mice, it could be interesting to know if there were differences between the results from male or female animals. If this was the case, the results should be separated and re-analyzed. One more factor (i.e., gender) should be added to the statistical analysis.
  2. In the last paragraph of the Introduction the authors must give more detailed information about the study design: what was the animal model used in the present study (mouse or rat), the type and the duration of the treatment, what parameters have been evaluated.

Reviewer 2 Report

The authors report on their study entitled:  Altered gut structure and anti-bacterial defense in adult mice treated with antibiotics during early life.

General comments to the authors:

The study presented here is focused on an important aspect of antibiotic treatments and its potential impact on the health of the recipient of the antibiotics. The study is well designed and the results are well presented. The number of samples used is statistically acceptable, except for the low sample numbers used to generate Figure 4 (gene expression analysis).

I disagree however with several conclusions made from the figures included in the manuscript. Specifically, it appears that conclusions were made based on statistical significance that might be artefactual. Of concern is also the fact that different statistical analysis was carried out for similar type of data. In dta presented as ±SEM, variation in data is large with significant error bar overlap. This becomes significant when the sample sizes are equal, or nearly equal. Examples:

Figure 1: Statistical analysis was performed by two-way ANOVA test with Sidak’s multiple comparisons test (b) or two-tailed unpaired t-test (c-h). Error bars indicate mean ± SD. Levels of significance are indicated (*p<0.05).

Conclusion (lines 95-98): Although the differences in intestinal permeability immediately after AB treatment are no longer present in adulthood, the increased weight and length of the SI suggest that early life AB have long-lasting effects on the intestine. (Disagree)

Figure 2: Statistical analysis was performed by Mann-Whitney test as data was not normally distributed when assessed by D’Agostino and Pearson normality test (b) or two-tailed unpaired t-test (c and e). Error bars indicate median with interquartile range (b) or mean ± SD (c and e). Levels of significance are indi-140 cated (*p<0.05).

Conclusion (lines 127-128): These results indicate that early life AB induce hyperproliferative crypts, especially in the proximal SI. (Disagree)

Figure 3: Statistical analysis was performed by one-tailed unpaired t-test or Mann-Whitney test, as data was not normally distributed when assessed by D’Agostino and Pearson normality test (Sct). Error bars indicate mean ± SD or median with interquartile range (Sct). Levels of significance are indicated (*p<0.05, **p<0.01).

Conclusion (lines 168-170): Overall, the adult SI recovers from the direct effects of early life AB treatment, with the exception of the greater number of Paneth cells, the higher relative expression of Sct, and the lower relative expression of Cck in the proximal SI. (Disagree)

Other comments:

(1) How many days is the lifetime of the epithelial cells in these animals? If shorter than the period between AB treatment and analysis, are the observed changes due to genetic changes in the stem cells that produce these cells or due to differences in hormone signalling that’s affecting the same cells exposed to AB?

(2) Authors did not assess the biodiversity of the microbiome in the intestines! If there are differences in the microbiome, then one may conclude that the observed changes are simply due to microbiome differences and not due to long term genetic and physiological changes in the tissue. In experimental set ups, animals normally are kept on a specific diet which does not allow for the natural recovery of the microbiome. As the pups were weaned on day P21 and after the AB treatment they are kept on a strict diet thereafter.

Figure 5D: exceptionally large variation in the control data (or inversely very small variation in the AB data) and this needs to be explained and its impact on the conclusion analyzed. It appears that the statistical significance obtained in this analysis is due to the unusual differences in the distribution of the data between the control and the AG groups. Note that the data for Saa3 is inconsistent with the data for Reg3b, Reg3g, Saa1 & Saa2 and it is the only gene where there is equivalent distribution/deviation of data points from the mean. I believe the conclusion from this data is based on a statistical artifact.    

Round 2

Reviewer 2 Report

Thank you for your responses. Although all of my concerns were addressed, perhaps point number 6 was misunderstood. My concern in this point was that in the experimental set up, the microbiome was not given a chance to recover to natural diversity, which in turn may allow the intestinal tissue to also recover from the AB treatments. Keeping the animals on a strict diet would be expected to restrict the microbiome diversity which does not occur under otherwise natural conditions whereby the animals are exposed to a wide spectrum of microbes, including beneficial bacteria. This is especially relevant if the conclusions are to be extended to humans. 

Point 6: Authors did not assess the biodiversity of the microbiome in the intestines! If there are differences in the microbiome, then one may conclude that the observed changes are simply due to microbiome differences and not due to long term genetic and physiological changes in the tissue. In experimental set ups, animals normally are kept on a specific diet which does not allow for the natural recovery of the microbiome. As the pups were weaned on day P21 and after the AB treatment they are kept on a strict diet thereafter.

Response 6: We do not exclude that the observed differences are associated with changes in the microbiome, as explained in the Discussion. Our results indicate that besides the previously shown impact of early life antibiotic treatment on the microbiome and intestinal immunity, also epithelial defense response is altered. We focused on differences in the small intestinal epithelium because this study follows a previous paper demonstrating the direct effects of early life antibiotics on small intestinal epithelial cells, independent of disturbances in the microbiome.